# Association between miRNA Target Sites and Incidence of Primary Osteoarthritis in Women from Volga-Ural Region of Russia: A Case-Control Study

**DOI:** 10.3390/diagnostics11071222

**Published:** 2021-07-06

**Authors:** Anton Tyurin, Daria Shapovalova, Halida Gantseva, Valentin Pavlov, Rita Khusainova

**Affiliations:** 1Internal Medicine Department, Bashkir State Medical University, 450008 Ufa, Russia; halida.ganceva@mail.ru; 2Laboratory of Human Molecular Genetics, Institute of Biochemistry and Genetics, 450000 Ufa, Russia; daria-ufa92@mail.ru; 3Urology Department, Bashkir State Medical University, 450008 Ufa, Russia; pavlov@bashgmu.ru; 4Medical Genetics Department, Bashkir State Medical University, 450008 Ufa, Russia; ritakh@mail.ru

**Keywords:** gene, locus, osteoarthritis, miRNA, ethnicity

## Abstract

Over the past decades, numerous studies on the genetic markers of osteoarthritis (OA) have been conducted. MiRNA targets sites are a promising new area of research. In this study, we analyzed the polymorphic variants in 3′ UTR regions of *COL1A1*, *COL11A1*, *ADAMTS5*, *MMP1*, *MMP13*, *SOX9*, *GDF5*, *FGF2*, *FGFR1*, and *FGFRL1* genes to examine the association between miRNA target site alteration and the incidence of OA in women from the Volga-Ural region of Russia using competitive allele-specific PCR. The T allele of the rs9659030 was associated with generalized OA (OR = 2.0), whereas the C allele of the rs229069 was associated with total OA (OR = 1.43). The T allele of the rs13317 was associated with the total OA (OR = 1.67). After Benjamini-Hochberg correction, only rs13317 remained statistically significant. According to ethnic heterogeneity, associations between the T allele (rs1061237) with OA in women of Russian descent (OR = 1.77), the G allele (rs6854081) in women of Tatar descent (OR = 4.78), the C allele (rs229069) and the T allele (rs73611720) in women of mixed descent and other ethnic groups (OR = 2.25 and OR = 3.02, respectively) were identified. All associations remained statistically significant after Benjamini-Hochberg correction. Together, this study identified miRNA target sites as a genetic marker for the development of OA in various ethnic groups.

## 1. Introduction

Osteoarthritis (OA) is one of the most common joint diseases, often causing persistent pain and immobilization [1,2]. The genetic contribution to OA varies from 40% to 65%, depending on the presence of the pathology in the immediate family, the location of the affected joint, sex, and ethnicity of the patients [3,4,5,6]. A large number of studies using both the gene-candidate approach and the genome-wide association study (GWAS) have been conducted, but the existing findings still fail to elucidate the molecular pathogenesis of the disease [7]. The study of miRNA target sites is a promising new direction in molecular diagnostics. MiRNAs are single-stranded non-coding RNAs that are about 19–25 bps long and inhibit protein expression by directly binding to the 3′ UTR region of target mRNAs. Recent studies have shown that miRNAs are also involved in the development and progression of OA [8]. A large-scale analysis of the expression of 365 miRNAs using microarrays revealed nine miRNAs (miR-16, miR-22, miR-23b, miR-30b, miR-103, miR-223, miR-377, miR-483, and miR-509) that were associated with upregulation of the genes potentially involved in cartilage homeostasis in patients with OA compared to normal cartilage, and seven miRNAs (miR-25, miR-26a, miR-29a, miR-140, miR-210, miR-337, and miR-373) that were associated with the downregulation of the *PPARA*, *BMP7*, and *MMP13* genes [9]. Recent studies of healthy cartilages and cartilages with knee OA have shown that miR-125b suppresses aggrecanase gene expression (*ADAMTS4*) and IL-1β activation in cartilage with OA [10]. Park et al. demonstrated that miR-558 reduces the cytochrome-C oxidase-2 (*COX2*) gene expression through the activation of IL-1β and stimulates the catabolism of chondrocytes in OA [11]. Even small variations in the 3′ UTR of the target sequence can alter the miRNA binding, leading to changes in these key regulatory interactions [12], so that even a single nucleotide difference can have profound effects [13]. Aberrations of miRNA expression patterns have been shown to be involved in oncogenesis, systemic diseases of connective tissue, osteoporosis, and a number of other conditions [14,15,16]. As more and more miRNAs are revealed to play a role in chondrogenesis, cartilage homeostasis, and OA pathogenesis, it is imperative to identify the key miRNAs and their targets that could potentially affect the disease progression. The aim of our work was to study the microRNA binding sites in patients with OA of different localizations in various ethnic groups of the Volga-Ural region of Russia.

## 2. Materials and Methods

Patient samples. A total of 417 women (mean age 51.67 ± 11.5) from inpatient care units of Ufa (Republic of Bashkortostan, Russia) hospitals participated in the study between January 2013 and August 2017. The patients were examined to diagnose for osteoarthritis according to the criteria of the American College of Rheumatologists (1995) with X-ray confirmation. Overall, 356 women with OA were recruited and divided into three groups as follows: (1) subgroup 1 included 84 women affected by generalized OA, (2) subgroup 2 included 197 women affected by knee OA, and (3) subgroup 3 included 75 women affected by hip OA. Patients with oncological pathology, systemic diseases of the connective tissue (rheumatoid arthritis, systemic lupus erythematosus, etc.), signs of an active inflammatory process of both infectious and non-infectious etiology, or traumatic joint injuries in the anamnesis were excluded from the study. A total of 161 normal healthy individuals were recruited to form the age-matched control group. The protocol was approved by the Ethics Review Committee of the Bashkir State Medical University, and signed informed consent was obtained from all of the participants. The ethnic composition of the sample was as follows: 144 Russian (Slavic group of the Indo-European language family), 159 Tatar (Turkic branch of the Altaic language family), and 114 mixed and representatives of small ethnic groups (Table 1).

Patient and public involvement participants were not involved in designing the research question, conducting the study, or in the interpretation or writing of the results. There are no plans to involve participants or relevant patient communities in dissemination of results. Results are not disseminated to study participants.

DNA extraction and genotyping. Genomic DNA was extracted from peripheral blood leucocytes by phenol-chloroform extraction (Mathew, 1984). Genotyping was performed using a competitive allele-specific PCR (KASP)—patented technology by LGC-Genomics. Selection of the miRNA target loci was carried out using the database of the National Center for Biotechnological Information (https://www.ncbi.nlm.nih.gov/, accessed on 1 November 2020), Ensemble GenomeBrowser (www.ensembl.org, accessed on 1 November 2020) and the base of polymorphisms of microRNA target sites (http://compbio.uthsc.edu/miRSNP, accessed on 1 November 2020). The candidate gene selection was based on the involvement in connective tissue metabolism or disease development (Table 2).

Studies of their polymorphic variants in the coding part do not fully explain the development of pathology [17,18,19,20,21]. The genotyping results were obtained using QuantStudio 12K FlexReal-Time PCR System (Thermo Fisher Scientific, Waltham, MA, USA). Genotyping was performed by the laboratory staff who were “blind” to the status of the samples. Further, 10% of the samples were tested twice to validate the genotyping results with 100% reproducibility. Two authors independently reviewed the genotyping results, data entry, and statistical analyses.

Statistical analysis. A standard χ^2^-analysis was used to examine the differences in allelic frequencies and genotype distributions between the patients with OA and controls using the Statistica software version 6.0 (http://statsoft.ru/, accessed on 1 November 2020). Considering the type I error caused by multiple testing, *p* values were adjusted by calculating the FDR value using the Benjamini-Hochberg method (https://tools.carbocation.com/FDR, accessed on 1 November 2020). Hardy-Weinberg equilibrium and the heterozygosity of the studied loci were estimated using the Haploview 2.0. The OR and 95% CI were calculated using the reported risk allele as a reference. Statistical significance was considered at *p* < 0.05.

## 3. Results

### 3.1. Genetic Association Analysis

A total of 12 polymorphic loci in the 3′ UTR region of 17 genes, involved in the formation and degradation of connective tissue, were genotyped. Hardy-Weinberg equilibrium was maintained for all the loci except for rs9978597. When considering the control sample independently, the indicator corresponded to equilibrium values (Table 2). The distributions of alleles and genotypes for the selected loci are available from the Appendix A.

The T allele of the rs9659030 (*COL11A1*) was associated with the generalized OA (*p* = 0.019; OR = 2.0; 95% CI 1.11 to 3.62), the TT genotype was associated with total patients (*p* = 0.026; OR = 1.59; 95% CI 1.05 to 2.42), generalized OA (*p* = 0.003; OR = 2.75; 95% CI 1.39 to 5.46), and OA of the hip (*p* = 0.016; OR = 2.3; 95% CI 1.14 to 4.36). With respect to rs229069 (*ADAMTS5*), a significant association was found between the C allele and the incidence of total OA (*p* = 0.018; OR = 1.43; 95% CI 1.06 to 1.93), as well as with knee OA (*p* = 0.042; OR = 1.43; 95% CI 1.01 to 2.03) and hip OA (*p* = 0.026; OR = 2.039; 95% CI 1.08 to 3.85). The CC genotype was also found to be associated with total OA (*p* = 0.037; OR = 1.53; 95% CI 1.02 to 2.28) and hip OA (*p* = 0.026; OR = 2.039; 95% CI 1.08 to 3.85). The T allele of rs13317 (*FGFR1*) was strongly associated with the total OA (*p* = 0.001; *p** = 0.01; OR = 1.67; 95% CI 1.2 to 2.3), knee OA (*p* = 0.003; *p** = 0.03; OR = 1.74; 95% CI 1.19 to 2.55), and generalized OA (*p* = 0.044; OR = 1.67; 95% CI 1.01 to 2.75) (Table 3).

After Benjamini-Hochberg correction, rs13317 remained statistically significant in the following groups: the T allele in control vs. total patients (adjusted *p** = 0.01) and in control vs. OA of the knee (adjusted *p** = 0.03). Neither the genotype nor the allele frequencies of rs1061347 (*COL1A1*), rs229077, rs9978597 (*ADAMTS5*), rs5854, rs470215 (*MMP1*), rs1042840 (*MMP13*), rs1042673 (*SOX9*), rs4647940 (*FGFRL1*) were significantly different between the affected individuals and normal controls.

### 3.2. Ethnic Analysis

Due to the profound ethnic heterogeneity of residents of the Volga-Ural region of Russia [22], we analyzed the distribution of alleles and genotypes in Russian, Tatar, and mixed individuals combined with small ethnic groups, which included Bashkir, Chuvash, Kabardian, Mari, Armenian, Azerbaijani, Belarusian, and Ossetian. The following associations between a given allele and incidence of total OA in women of a certain descent were identified: T allele (rs1061237) was enriched in women of Russian descent (OR = 1.77; 95% CI 1.07 to 2.94), the G allele (rs6854081) was enriched in Tatar women (OR = 4.78; 95% CI 1.89 to 12.02), the C allele (rs229069), and T allele (rs73611720) prevailed in the groups of mixed ancestry and small ethnic groups combined (OR = 2.25; 95% CI 1.30 to 3.89 and OR = 3.02; 95% CI 1.38 to 6.60, respectively) (Table 4). All associations remained statistically significant after Benjamini-Hochberg correction.

Subsequently, we performed meta-analysis to verify the collected data and identified general and ethnospecific trends (Table 5).

In rs6854081 and rs4836732 loci, high genetic heterogeneity of the studied sample was found (I^2^ = 79.81 and 77.43, respectively). These loci did not produce statistically significant values in regard to the incidence of the total OA in women but were identified as ethnically specific markers of OA progression in women of Tatar descent. High heterogeneity was also inherent in rs7639618, rs1061237, and rs9350591 loci, which were distinguished as the markers of the OA progression in women of Russian descent. However, these associations did not remain statistically significant after adjusting for multiple comparisons. The association of the rs229069 is noteworthy. It was associated with total OA and, according to the high ethnic specificity and genetic heterogeneity, with total OA only in mixed and other ethnic groups at the same time. Most likely, the rs229069 is an ethnospecific marker in Bashkir women, since the number of women of Bashkir descent prevailed in this cohort. The obtained associations must be confirmed in future studies with a larger sample size and a cohort restricted to Bashkir women exclusively.

## 4. Discussion

It is generally recognized that OA is a complex, multi-factorial disease and genetic factors have been suggested to play an important role in the development and progression of cartilage degeneration. More than 50 candidate genes related to OA have been identified by genetic association studies. Unfortunately, today, science is faced with a problem of missing heritability. One way to solve this problem is to study epigenetic regulatory mechanisms, including miRNAs and their target sites. However, the most common weakness is difficulty in replicating the previous association signals. Furthermore, studies of microRNA binding sites to date are not very numerous.

### 4.1. rs9659030 (COL11A1)

When the A variant is present in the seed site, miR-7-1-3p and miR-7-2-3p fail to bind to the respective genes, whereas the presence of the G variant in the seed site creates new binding sites for miR-495-3p and miR-5688, respectively. The role of these miRNAs in the pathogenesis have not been examined yet, but these observations are, nevertheless, compelling. A recent EVA study by Raine et al. analyzed the allelic expression imbalance (AEI) of the *COL11A1* gene using three polymorphic loci: rs9659030, rs1676486, and rs2615977. In this study, a decrease in the expression of the *COL11A1* gene was detected in the presence of a rare T allele at the polymorphic locus rs1676486 (p.Ser1419Thr), which has been shown to be in linkage disequilibrium with rs9659030. AEI at rs1676486 locus is a risk factor for lumbar disc herniation (LDH), but not for OA. Therefore, this suggests that the two diseases share a common functional phenotype, namely, AEI of *COL11A1*, but this appears to be a disease risk only in LDH [23]. Another noteworthy study of allelic imbalance (AI) was done by den Hollander et al. Their findings were based on the AI effect observed in individuals heterozygous for both rs2615977 and rs9659030 loci. Two independent *COL11A1* SNPs (D′ = 0.2, r^2^ = 0.01) with profoundly significant meta-φ AI were revealed: the former was the abovementioned rs9659030 locus with meta-φ AI of 0.65 (P = 1.2 × 10−25) and the rs2229783 locus with meta-φ AI of 0.53 (P = 3.7 × 10−8) was the latter. Notably, in contrast to the results obtained by Raine et al., the extent and consistency of AI for rs9659030 were considerably higher than those for rs2229783 and rs1676486 loci [24]. There were no statistically significant associations at rs9659030 locus in regard to the development of glaucoma and myopia in the Chinese population [25,26]. However, further studies are required to confirm these findings.

### 4.2. rs229069 (ADAMTS5)

Unfortunately, limited literature is available on this polymorphic variant. The G allele disrupts the interaction of miR-3144-3p and miR-875-5p with the target mRNA, whereas the C allele creates a novel binding site for let-7d-3p and let-7e-3p miRNAs, which suggests a change in the regulation of expression of the *ADAMTS5* gene. Our findings regarding the association of this locus with the development of OA are a significant contribution to the understanding of the molecular pathogenesis of the disease.

### 4.3. rs13317 (FGFR1)

The A allele at 3′ UTR of rs13317 harbors binding sites for both miR-3128 and miR-4470, while miRNAs associated with the G allele at this locus are not currently defined. However, this polymorphic variant was found to be associated with the pathologies of the nervous system and bone tissue development; an association with the anomalies in the development of the facial skeleton (overall head size and midfacial development, the ratio of the distance between the eyes) was observed in patients from Japan and Korea [27]. The CC genotype occurred 50% as often in patients with ossification of the posterior longitudinal ligament, as compared to the control group in Koreans [28]. In the study of 203 patients with an injury of the rotator cuff of the shoulder, an association between the T allele and the presence of pathology was revealed (P = 0.08/0.02; OR 2.67; CC overrepresented in control patients) [29]. In a study of non-union (NU) fractures, 167 patients with long bone fractures, 101 with uneventful healing, and 66 presenting aseptic non-unions were investigated. A significant association of *FGFR1* (rs13317) C allele with NU was observed. Carrying the C allele increased twice the risk of NU, in contrast to TT genotype, which acts as a protective factor [30].

### 4.4. rs6854081 (FGF2)

Fibroblast growth factor is one of the key links in bone metabolism. It is expressed in most mesenchymal and bone-related cells, including chondrocytes, osteoblasts, adipocytes and osteoloclasts. *FGF2* is expressed in the embryos of the extremities at the developmental stage and controls the growth and formation of limb patterns. In bone cells, by regulating bone formation induced by parathyroid hormone (PTH) and bone morphogenetic protein 2 (BMP2) promotes the differentiation of bone marrow stromal cells into osteoblast [31,32,33]. It was found that *rs6854081* and *rs1048201* in the 3′-UTR of FGF2 are associated with bone mineral density (BMD). The minor *G* allele of *rs6854081* was associated with low BMD in a cohort of white women (*n* = 2725). The study demonstrated that the *FGF2* transcript containing the *T* allele of *rs6854081* had a lower expression level in monocytes and B cells, and the allele was associated with higher BMD. *T* allele could have a higher binding affinity for miR-146a/b, therefore, inhibiting *FGF2* translation. Suppression of *FGF2* will reduce osteoclastogenesis, which in turn will lead to an increase in BMD [34]. In another study, the minor *T* allele of *rs1048201* was significantly associated with spinal BMD in the Han people (*n* = 2339), it may act by affecting binding of hsa-miR-196a-3p [35]. Study of the *rs6854081*, *rs1048201* and *rs7683093* of *FGF2* in one more Chinese ethnic cohort—Zhuang, revealed significant relationship of these loci with the bone density of of the femoral neck [36]. However, a study only by the Asian could not reproduce the relationship between target loci and BMD, suggesting that the effects of *rs6854081*, *rs1048201*, *rs7683093* on bone regulation may be ethnic specific [37]. In this regard, results of our study on associations of the *rs6554081* in particular ethnic groups are especially important.

### 4.5. rs1061237 (COL1A1)

There is little research on the pathogenetic significance of this polymorphic variant. In 276 patients from the USA with severe myopia (−9.5 D), an association with this locus was revealed, but it did not remain statistically significant after correction for the multiplicity of comparisons [38]. There was a strong statistical correlation between the C allele at rs1061237 locus and cutaneous leishmaniasis in patients from Corte de Pedra, Bahia, Brazil [39]. In another study, a team of specialists from the Chinese Academy of Medical Sciences Cancer Hospital investigated this locus to assess the risk of lung cancer progression in 430 patients, but no statistically significant results were obtained [40]. The T allele at rs1061237 locus can potentially have many interactions with a number of miRNAs: miR-1226-5p, miR-1304-5p, miR-1914-3p, miR-3184-5p, miR-423-5p, miR-5194, miR-6732 -5p, miR-6734-5p, miR-6738-5p, miR-6834-5p, or miR-8055, whereas the C allele for can potentially interact with miR-1275, miR-328-5p, miR-4260, miR-450a-2-3p, miR-4665-5p, miR-5572, miR-6751-5p, miR-6795-5p, miR-6803-5p, miR-6885-5p, miR-6887-5p, and miR-7109-5p.

### 4.6. Limitations of the Study

A limitation is that the sample size was relatively small and the study was conducted only among female patients; polymorphic target sites were not evaluated for the level of microRNA expression.

## 5. Conclusions

The current study demonstrated an association between the T allele of rs13317 in *FGFR1* and incidence of the total OA and knee OA in women from Volga-Ural region of Russia. The G allele of rs6854081 in *FGF2* was a risk factor for OA development for women of Tatar descent, the C allele of rs1061237 in *COL1A1* was a risk factor for OA development for Russian women, and the T allele of rs229069 in *ADAMTS5* and the T allele of rs73611720 in *GDF5* were risk factors for OA development for women of mixed descent.

## Figures and Tables

**Table 1 diagnostics-11-01222-t001:** Characteristics of the study subjects.

	Control *n* = 161	Total OA *n* = 256	Generalized OA *n* = 61	Knee OA *n* = 139	Hip OA *n* = 56
Age (years)	45.55 ± 12.55	55.67 ± 11.50	52.44 ± 9.02	57.13 ± 7.98	55.44 ± 8.64
Body mass index, kg/m^2^	25.24 ± 7.34	29.98 ± 5.41	28.57 ± 5.10	30.49 ± 5.51	30.35 ± 5.39
Sex	female	female	female	female	female
Ethnicity, %	Russian, 30.43 Tatar, 42.86 Mixed, 26.71	Russian, 37.11 Tatar, 35.16 Mixed, 27.73	Russian, 24.59 Tatar, 39.34 Mixed, 36.07	Russian, 38.13 Tatar, 37.41 Mixed, 24.46	Russian, 48.21 Tatar, 25.00 Mixed, 26.79

OA—osteoarthritis.

**Table 2 diagnostics-11-01222-t002:** Characterization of the studied loci in miRNA target sites.

№	SNP	Position	Gene, Loci	Allele	miR ID	Function Class	Genotyped, %	HW_pval_
**1**	rs9659030	c.*1183A > G	*COL11A1* (1p21.1)	*A	hsa-miR-495-3p hsa-miR-5688	D	93.53	0.287
				*G	hsa-miR-7-1-3p hsa-miR-7-2-3p	C		
**2**	rs1061237	c.*88T > C	*COL1A1* (1p21.1)	*T	hsa-miR-1226-5p hsa-miR-1304-5p hsa-miR-1914-3p hsa-miR-3184-5p hsa-miR-423-5p hsa-miR-5194 hsa-miR-6732-5p hsa-miR-6734-5p hsa-miR-6738-5p hsa-miR-6834-5p hsa-miR-8055	C	97.84	0.802
				*C	hsa-miR-1275 hsa-miR-328-5p hsa-miR-4260 hsa-miR-450a-2-3p hsa-miR-4665-5p hsa-miR-5572 hsa-miR-6751-5p hsa-miR-6795-5p hsa-miR-6803-5p hsa-miR-6885-5p hsa-miR-6887-5p hsa-miR-7109-5p	N		
**3**	rs1061947	c.*744C > T		*C	hsa-miR-4258 hsa-miR-7108-3p	D	97.60	0.183
				*T	hsa-miR-1224-3p hsa-miR-1260a hsa-miR-1260b hsa-miR-150-5p hsa-miR-2116-3p hsa-miR-4713-5p hsa-miR-500b-3p hsa-miR-532-3p	C		
**4**	rs229077	c.*1033A > G	*ADAMTS5* (21q21.3)	*A	hsa-miR-105-5p hsa-miR-4719 hsa-miR-5700 hsa-miR-586 hsa-miR-7853-5p	O	100.00	0.018
				*G	hsa-miR-3912-3p	O		
**5**	rs229069	c.*5689G > T		*G	hsa-miR-3144-3p hsa-miR-875-5p	D	97.60	0.682
				*T	hsa-let-7d-3p hsa-let-7e-3p	C		
**6**	rs9978597	c.*2229A > C		*A	hsa-miR-15b-3p hsa-miR-494-3p	O	96.40	0.526
**7**	rs5854	c.*269C > T	*MMP1* (11q22.2)	*C	hsa-miR-1299 hsa-miR-3148 hsa-miR-3664-5p hsa-miR-4714-5p hsa-miR-514a-5p hsa-miR-518c-5p hsa-miR-6124 hsa-miR-6128 hsa-miR-875-3p	D	99.04	0.383
				*T	hsa-miR-517-5p hsa-miR-5684 hsa-miR-6508-5p hsa-miR-8067	C		
**8**	rs470215	c.*44A > T		*A	hsa-miR-1248 hsa-miR-6844 hsa-miR-9-5p	D	85.37	1.0
				*T	hsa-miR-1306-5p hsa-miR-3121-5p	C		
**9**	rs1042840	c.*885A > G	*MMP13* (11q22.2)	*A	hsa-miR-3654 hsa-miR-4638-3p	D	91.13	0.955
				*G	hsa-miR-3687 hsa-miR-4442	C		
**10**	rs73611720	c.*335A > C	*GDF5* (20q11.22)	*A	hsa-miR-6770-5p hsa-miR-1253 hsa-miR-4251	N	99.04	1.0
					hsa-miR-3929 hsa-miR-4419b hsa-miR-4478 hsa-miR-485-5p hsa-miR-6884-5p	D		
				*C	hsa-miR-25-5p hsa-miR-4434 hsa-miR-4516 hsa-miR-5703 hsa-miR-6087	C		
**11**	rs1042673	c.*811A > G	*SOX9* (17q24)	*A	hsa-miR-190a-3p hsa-miR-5011-5p	C	99.76	0.578
				*G	hsa-let-7d-3p	D		
**12**	rs13317	c.*1632A > G	*FGFR1* (8p11.23)	*A	hsa-miR-3128 hsa-miR-4470	C	99.28	0.809
**13**	rs4647940	c.*1256C > G	*FGFRL1* (4p16.3)	*C	hsa-miR-1296-3p hsa-miR-194-3p hsa-miR-491-5p hsa-miR-6499-3p	N	99.52	1.0
				*G	hsa-miR-4667-5p hsa-miR-4700-5p hsa-miR-8089	C		
**14**	rs6854081	c.*3156T > A	*FGF2* (4q28.1)	*T	hsa-miR-146a-5p hsa-miR-146b-5p hsa-miR-3919 hsa-miR-3925-5p hsa-miR-7153-5p	C	95.44	0.125
				*A	hsa-miR-6858-3p	N		

C—creates a new miRNA target site, D—disrupts a conserved miRNA target site, N—disrupts a non-conserved miRNA target site, O—function is unknown.

**Table 3 diagnostics-11-01222-t003:** Associations of the miRNA target sites loci in different OA localization subgroups.

SNP	Localisation	Allele	*p*	*p**	OR; 95% CI
*rs9659030*	Total OA	TT	0.026	0.337	OR = 1.59; (1.05–2.42)
	Generalized OA	T	0.019	0.170	OR = 2.0; (1.11–3.62)
	Hip OA	TT	0.016	0.208	OR = 2.3; (1.14–4.36)
*rs229069*	Total OA	C	0.018	0.161	OR = 1.43; (1.06–1.93)
	Hip OA	C	0.026	0.233	OR = 1.63; (1–2.66)
	Knee OA	C	0.042	0.378	OR = 1.43; (1.01–2.03)
*rs13317*	Total OA	T	0.001	0.01	OR = 1.67; (1.2–2.3)
	Knee OA	T	0.003	0.03	OR = 1.74; (1.19–2.55)
	Generalized OA	T	0.044	0.395	OR = 1.67; (1.01–2.75)

OA—osteoarthritis.

**Table 4 diagnostics-11-01222-t004:** Associations of the miRNA target sites loci in different ethnic groups.

SNP	Ethnicity	Allele	*p*	*p**	OR; 95% CI
*rs6854081*	Tatar	G	0.0001	0.0002	OR = 4.78; (1.89–12.02)
*rs1061237*	Russian	C	0.017	0.034	OR = 1.77; (1.07–2.94)
*rs229069*	Mixed	T	0.0002	0.0004	OR = 2.25; (1.30–3.89)
*rs73611720*	Mixed	T	0.004	0.008	OR = 3.02; (1.38–6.60)

**Table 5 diagnostics-11-01222-t005:** Meta-analysis of microRNA target sites loci associations in patients with and without OA.

SNP	Fixed Effect Model		Random Effect Model		Q	I^2^
	P	OR	P (R)	OR (R)		
rs9659030	0.230	0.809	0.230	0.809	0.719	0.00
rs4647940	0.997	0.999	0.999	1.195	0.811	0.00
rs6854081	0.552	1.152	0.738	1.120	0.120	52.43
rs1057972	0.725	1.058	0.779	1.065	0.143	48.51
rs1054204	0.094	1.272	0.107	1.272	0.341	6.94
rs13317	0.003	0.607	0.003	0.606	0.353	3.89
rs3128575	0.383	1.155	0.383	1.155	0.739	0.00
rs10793442	0.35	0.831	0.35	0.831	0.989	0.00
rs5854	0.834	0.968	0.834	0.968	0.453	0.00
rs2239008	0.157	1.319	0.193	1.326	0.297	17.56
rs470215	0.763	0.950	0.743	0.925	0.146	48.10
rs1042840	0.225	0.818	0.371	0.829	0.208	36.35
rs2463018	0.918	1.017	0.918	1.017	0.520	0.00
rs1061947	0.158	0.772	0.158	0.772	0.945	0.00
rs1061237	0.614	0.927	0.902	0.959	0.006	79.99
rs1042673	0.231	0.842	0.231	0.842	0.507	0.00
rs73611720	0.286	0.780	0.628	0.802	0.024	72.97
rs9509	0.428	0.725	0.428	0.725	0.465	0.00
rs229069	0.015	20.68	0.249	0.670	0.0073	79.67
rs9978597	0.230	1.528	0.230	1.528	0.632	0.00
rs229077	0.700	1.057	0.771	1.055	0.202	37.43

P—*p*-value fixed; P(R)—*p*-value random; Q—Cohren heterogeneity criterion; I^2^—Higgins heterogeneity criterion.

## Data Availability

The additional data are available in Appendix A.

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
