# Peer review of "Association between miRNA Target Sites and Incidence of Primary Osteoarthritis in Women from Volga-Ural Region of Russia: A Case-Control Study"

_diagnostics, 2021, doi:10.3390/diagnostics11071222_

Round 1

Reviewer 1 Report

The authors studied the association between miRNA target sites and incidence of primary OA and demonstrated that association between alleles of SNPs in FGFR1 with incidence of OA, and FGF2, ADAMTS5, COL1A1, and GDF5 as OA risk factors of OA development in women of different descent.

Comments

  1. Conclusions should not be duplicated. This should be corrected.
  2. Line 241: rs13317 belongs to FGFR1. This should be corrected.
  3. It is not clear why SNPs in COL1A1 and GDF5 as a risk factors for OA development in Russian and Mixed descent, respectively, were not mentioned in the Conclusions? This should be clarified.
  4. It is not clear why the authors did not discuss SNPs related to FGF2 and GDF5 genes, which appeared to be important as a risk factors of OA development. This should be clarified.

Reviewer 2 Report

This work by Tyurin et al. entitled “Association between miRNA target sites and incidence of primary osteoarthritis in women from Volga-Ural region of Russia: a case-control study” explores the association of polymorphic variants in the 3’-UTR of a number of genes and OA in female individuals form a specific geographic region in Russia. The authors report significant implications in three genetic loci which are associated with the ethnic background in this cohort.

The idea is very interesting and the manuscript is well written. However, some concerns arise after careful consideration, that need to be addressed.

 Major concerns:

  1. The authors state that they identified biomarkers for OA. To claim that, additional and more detailed analyses should be performed with a larger cohort and appropriate staging of the disease severity. I would suggest to rephrase the corresponding sentences in the text.

  1. The introduction is well written but there is no clear aim of the study. Why the authors selected this particular population and what are the specific characteristics that these individuals exhibit and need to be further explored? Please provide more details. Is there any epidemiological study for this specific region with regards to OA prevalence?

  1. Furthermore, there in not a sufficient justification of the selection of the specific genetic loci. For example, why did the authors not examine other important OA-related genes, e.g. COL2A1, ADAMTS4, ACAN etc.

  1. Since these genetic loci are described in the context of miRNA target sites, it is important to include the specific miRNAs for every target, for example as an additional column in Table 2. Do the different variants affect the binding capacity of these miRNAs?

  1. It would be extremely useful and would greatly add to this work if the authors are in position to report the levels of some of these miRNAs either in circulation or in the synovial fluid of the patients

Minor:

  1. Lines 15-16. Is there any particular reason for not including GDF5 and FGF2 variants in the abstract?
  2. Line 17, competitive.
  3. Line 34-35, please correct the GWAS abbreviation.
  4. Line 46, which are the above mentioned genes?
  5. Line 47-48, please rephrase. As it stands, it leads to the conclusion the IL1β activation suppresses ADAMTS4.
  6. Lines 67-68, “systemic diseases of the connective tissue”. Please write in parentheses one or two examples.
  7. Line 84, competitive.
  8. Line 198, replace little with limited.
  9. Line 241, is the T allele of rs13317 in FGFR2 or FGFR1?
  10. Please remove the double numbering from all citations.

Round 2

Reviewer 2 Report

All points sufficiently addressed by the authors. The manuscript has been substantially improved.

Author Response

We express our deep gratitude to the reviewer for the work done and help in improving the publication